# Calibrated Seq2seq Models for Efficient and Generalizable Ultra-fine Entity Typing

**Yanlin Feng** [†][§][*]   **Adithya Pratapa**[†]   **David Mortensen**[†]

[†]Language Technologies Institute, Carnegie Mellon University
[§]Megagon Labs
yanlin@megagon.ai, {vpratapa, dmortens}@cs.cmu.edu

## Abstract

Ultra-fine entity typing plays a crucial role in information extraction by predicting fine-grained semantic types for entity mentions in text. However, this task poses significant challenges due to the massive number of entity types in the output space. The current state-of-the-art approaches, based on standard multi-label classifiers or cross-encoder models, suffer from poor generalization performance or inefficient inference. In this paper, we present CASENT, a seq2seq model designed for ultra-fine entity typing that predicts ultra-fine types with calibrated confidence scores. Our model takes an entity mention as input and employs constrained beam search to generate multiple types autoregressively. The raw sequence probabilities associated with the predicted types are then transformed into confidence scores using a novel calibration method. We conduct extensive experiments on the UFET dataset which contains over $10k$ types. Our method outperforms the previous state-of-the-art in terms of F1 score and calibration error, while achieving an inference speedup of over $50$ times. Additionally, we demonstrate the generalization capabilities of our model by evaluating it in zero-shot and few-shot settings on five specialized domain entity typing datasets that are unseen during training. Remarkably, our model outperforms large language models with 10 times more parameters in the zero-shot setting, and when fine-tuned on 50 examples, it significantly outperforms ChatGPT on all datasets.[1]

## 1 Introduction

Classifying entities mentioned in text into types, commonly known as entity typing, is a fundamental problem in information extraction. Earlier research on entity typing focused on relatively small

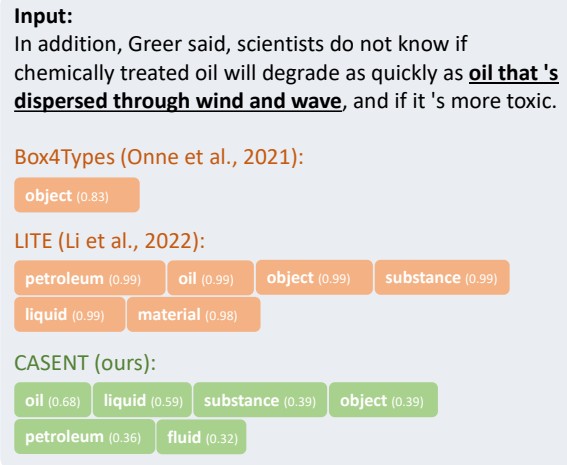

**Input:**
In addition, Greer said, scientists do not know if chemically treated oil will degrade as quickly as **oil that 's dispersed through wind and wave**, and if it 's more toxic.

Box4Types (Onne et al., 2021):
object (0.83)

LITE (Li et al., 2022):
petroleum (0.99)   oil (0.99)   object (0.99)   substance (0.99)
liquid (0.99)   material (0.98)

CASENT (ours):
oil (0.68)   liquid (0.59)   substance (0.39)   object (0.39)
petroleum (0.36)   fluid (0.32)

Figure 1: Comparison of predicted labels and confidence scores for a UFET test example using Box4Types (Onoe et al., 2021), LITE (Li et al., 2022), and our approach, CASENT. Predictions are sorted in descending order based on confidence. Box4Types fails to generalize to rare and unseen types, while LITE does not predict calibrated confidence scores and exhibits slow inference speed.

type inventories (Ling and Weld, 2012) which imposed severe limitations on the practical value of such systems, given the vast number of types in the real world. For example, WikiData, the current largest knowledge base in the world, records more than 2.7 million entity types[2]. As a result, a fully supervised approach will always be hampered by insufficient training data. Recently, Choi et al. (2018) introduced the task of ultra-fine entity typing (UFET), a multi-label entity classification task with over $10k$ fine-grained types. In this work, we make the first step towards building an efficient general-purpose entity typing model by leveraging the UFET dataset. Our model not only achieves state-of-the-art performance on UFET but also generalizes outside of the UFET type vocabulary. An

---

[*] This work was done while the first author was at Carnegie Mellon University.

[1]Our code, models and demo are available at https://github.com/yanlinf/CASENT.

[2]Estimated from the unique children in the *subclassOf (P279)* relations using the February 2023 Wikidata dump.

example prediction of our model is shown in Figure 1.

Ultra-fine entity typing can be viewed as a multi-label classification problem over an extensive label space. A standard approach to this task employs multi-label classifiers that map contextual representations of the input entity mention to scores using a linear transformation (Choi et al., 2018; Dai et al., 2021; Onoe et al., 2021). While this approach offers superior inference speeds, it ignores the type semantics by treating all types as integer indices and thus fails to generalize to unseen types. The current state-of-the-art approach (Li et al., 2022) reformulated entity typing as a textual entailment task. They presented a cross-encoder model that computes an entailment score between the entity mention and a candidate type. Despite its strong generalization capabilities, this approach is inefficient given the need to enumerate all $10k$ types in the UFET dataset.

Black-box large language models, such as GPT-3 and ChatGPT, have demonstrated impressive zero-shot and few-shot capabilities in a wide range of generation and understanding tasks (Brown et al., 2020; Ouyang et al., 2022). Yet, applying them to ultra-fine entity typing poses challenges due to the extensive label space and the context length limit of these models. For instance, Zhan et al. (2023) reported that GPT-3 with few-shot prompting does not perform well on a classification task with thousands of classes. Similar observations have been made in our experiments conducted on UFET.

In this work, we propose CASENT, a **Ca**librated **S**eq2Seq model for **En**tity **T**yping. CASENT predicts ultra-fine entity types with calibrated confidence scores using a seq2seq model (T5-large (Raffel et al., 2020)). Our approach offers several advantages compared to previous methods: (1) Standard maximum likelihood training without the need for negative sampling or sophisticated loss functions (2) Efficient inference through a single autoregressive decoding pass (3) Calibrated confidence scores that align with the expected accuracy of the predictions (4) Strong generalization performance to unseen domains and types. An illustration of our approach is provided in Figure 2.

While seq2seq formulation has been successfully applied to NLP tasks such as entity linking (De Cao et al., 2020, 2022), its application to ultra-fine entity typing remains non-trivial due to the multi-label prediction requirement. A simple adaptation would employ beam search to decode multiple types and use a probability threshold to select types. However, we show that this approach fails to achieve optimal performance as the raw conditional probabilities do not align with the true likelihood of the corresponding types. In this work, we propose to transform the raw probabilities into calibrated confidence scores that reflect the true likelihood of the decoded types. To this end, we extend Platt scaling (Platt et al., 1999), a standard technique for calibrating binary classifiers, to the multi-label setting. To mitigate the label sparsity issue in ultra-fine entity typing, we propose novel weight sharing and efficient approximation strategies. The ability to predict calibrated confidence scores not only impacts task performance but also provides a flexible means of adjusting the trade-off between precision and recall in real-world scenarios. For instance, in applications requiring high precision, predictions with lower confidence scores can be discarded.

We carry out extensive experiments on the UFET dataset and show that filtering decoded types based on calibrated confidence scores leads to state-of-the-art performance. Our method surpasses the previous methods in terms of both F1 score and calibration error while achieving an inference speedup of more than 50 times compared to cross-encoder methods. Furthermore, we evaluate the zero-shot and few-shot performance of our model on five specialized domains. Our model outperforms Flan-T5-XXL (Chung et al., 2022), an instruction-tuned large language model with 11 billion parameters in the zero-shot setting, and surpasses ChatGPT when fine-tuned on 50 examples.

## 2 Related Work

### 2.1 Fine-grained Entity Typing

Ling and Weld (2012) initiated efforts to recognize entities with labels beyond the small set of classes that is typically used in named entity recognition (NER) tasks. They proposed to formulate this task as a multi-label classification problem. More recently, Choi et al. (2018) extended this idea to ultra-fine entity typing and released the UFET dataset, expanding the task to include an open type vocabulary with over $10k$ classes. Interest in ultra-fine entity typing has continued to grow over the last few years. Some research efforts have focused on modeling label dependencies and type hierarchies, such as employing box embeddings (Onoe et al., 2021) and contrastive learning (Zuo et al., 2022).

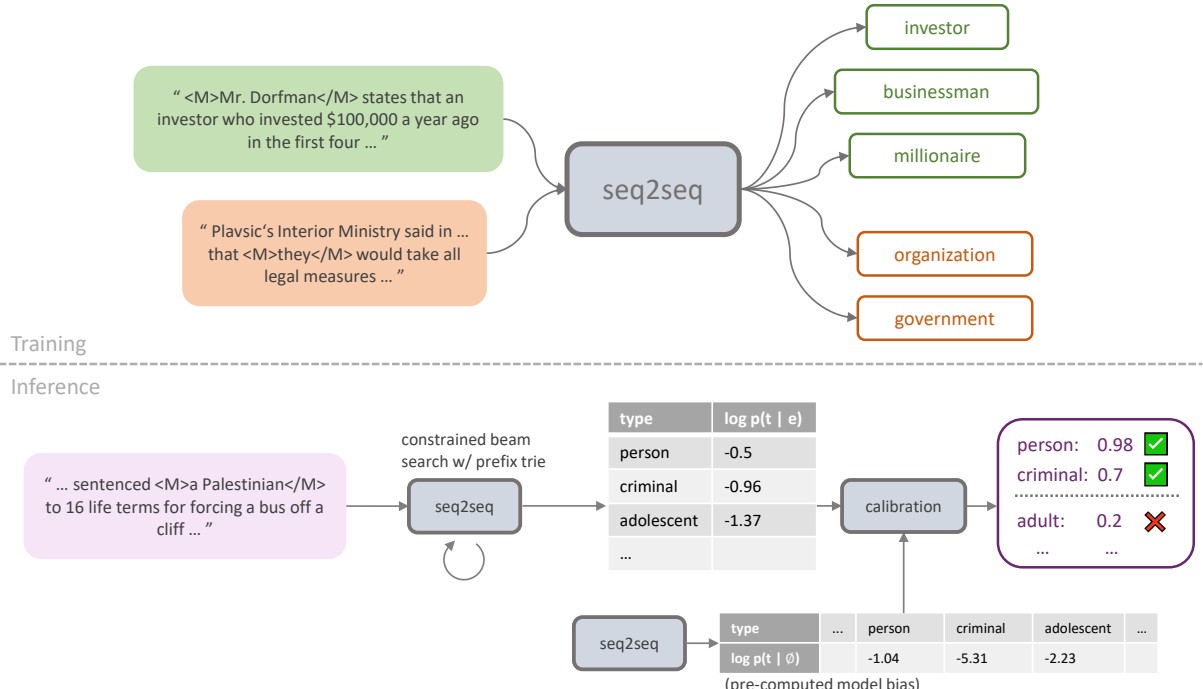

Figure 2: Overview of the training and inference process of CASENT. We present an example output from our model.

Another line of research has concentrated on data augmentation and leveraging distant supervision. For instance, Dai et al. (2021) obtained training data from a pretrained masked language model, while Zhang et al. (2022) proposed a denoising method based on an explicit noise model. Li et al. (2022) formulated the task as a natural language inference (NLI) problem with the hypothesis being an "is-a" statement. Their approach achieved state-of-the-art performance on the UFET dataset and exhibited strong generalization to unseen types, but is inefficient at inference due to the need to enumerate the entire type vocabulary.

## 2.2 Probability Calibration

Probability calibration is the task of adjusting the confidence scores of a machine learning model to better align with the true correctness likelihood. Calibration is crucial for applications that require interpretability and reliability, such as medical diagnoses. Previous research has shown that modern neural networks while achieving good task performance, are often poorly calibrated (Guo et al., 2017; Zhao et al., 2021). One common technique for calibration in binary classification tasks is Platt scaling (Platt et al., 1999), which fits a logistic regression model on the original probabilities. Guo et al. (2017) proposed temperature scaling as an extension of Platt scaling in the multi-class setting.

Although probability calibration has been extensively studied for single-label classification tasks (Jiang et al., 2020; Kadavath et al., 2022), it has rarely been explored in the context of fine-grained entity typing which is a multi-label classification task. To the best of our knowledge, the only exception is Onoe et al. (2021), where the authors applied temperature scaling to a BERT-based model trained on the UFET dataset and demonstrated that the resulting model was reasonably well-calibrated.

## 3 Methodology

In this section, we present CASENT, a calibrated seq2seq model designed for ultra-fine entity typing. We start with the task description (§3.1) followed by an overview of the CASENT architecture (§3.2). While the focus of this paper is on the task of entity typing, our model can be easily adapted to other multi-label classification tasks.

## 3.1 Task Definition

Given an entity mention $e$, we aim to predict a set of semantic types $\mathbf{t} = \{t_1, \ldots, t_n\} \subset \mathcal{T}$, where $\mathcal{T}$ is a predefined type vocabulary ($|\mathcal{T}| = 10331$ for the UFET dataset). We assume each type in the vocabulary is a noun phrase that can be represented by a sequence of tokens $t = (y_1, y_2, \ldots, y_k)$. We assume the availability of a training set $\mathcal{D}_{\text{train}}$ with annotated $(e, \mathbf{t})$ pairs as well as a development set

for estimating hyperparameters.

## 3.2 Overview of CASENT

Figure 2 provides an overview of our system. It consists of a seq2seq model and a calibration module. At training time, we train the seq2seq to output a ground truth type given an input entity mention by maximizing the length-normalized log-likelihood using an autoregressive formulation

$$\log p_\theta(t \mid e) = \frac{1}{k} \sum_{i=1}^{k} \log p_\theta(y_i \mid y_{<i}, e) \quad (1)$$

where $\theta$ denotes the parameters of the seq2seq model.

During inference, our model takes an entity mention $e$ as input and generates a small set of candidate types autoregressively via constrained beam search by using a relatively large beam size. We then employ a calibration module to transform the raw conditional probabilities (Equation 1) associated with each candidate type into calibrated confidence scores $\hat{p}(t \mid e) \in [0, 1]$.[3] The candidate types whose scores surpass a global threshold are selected as the model's predictions.

The parameters of the calibration module and the threshold are estimated on the development set before each inference run (which takes place either at the end of each epoch or when the training is complete). The detailed process of estimating calibration parameters is discussed in §3.4.

## 3.3 Training

Our seq2seq model is trained to output a type $t$ given an input entity mention $e$. In the training set, each annotated example $(e, \mathbf{t}) \in \mathcal{D}_{\text{train}}$ with $|\mathbf{t}| = n$ ground truth types is considered as $n$ separate input-output pairs for the seq2seq model.[4] We initialize our model with a pretrained seq2seq language model, T5 (Raffel et al., 2020), and finetune it using standard maximum likelihood objective:

$$\min_\theta \left[ - \sum_{(e,\mathbf{t}) \in \mathcal{D}_{\text{train}}} \sum_{t \in \mathbf{t}} \log p_\theta(t \mid e) \right] \quad (2)$$

Our seq2seq formulation greatly simplifies the training process by eliminating the need for nega-

[3]Here, we make a slight abuse of notation by treating $t$ as a binary random variable that indicates whether $e$ belongs to type $t$.

[4]Note that although a training example $(e, \mathbf{t})$ is separated into $n$ input-output pairs, the forward pass at the encoder only needs to be computed once.

tive sampling, which is required by previous cross-encoder approaches (Li et al., 2022; Dai et al., 2021).

## 3.4 Calibration

At the core of our approach is a calibration module that transforms raw conditional log-probability $\log p_\theta(t \mid e)$ into calibrated confidence $\hat{p}(t \mid e)$. We will show in section 4 that directly applying thresholding using $p_\theta(t \mid e)$ is suboptimal as it models the distribution over target token sequences instead of the likelihood of $e$ belonging to a certain type $t$. Our approach builds on Platt scaling (Platt et al., 1999) with three proposed extensions specifically tailored for the ultra-fine entity typing task: 1) incorporating model bias $p_\theta(t \mid \varnothing)$, 2) frequency-based weight sharing across types, and 3) efficient parameter estimation with sparse approximation.

**Platt Scaling**: We first consider calibration for each type $t$ separately, in which case the task reduces to a binary classification problem. A standard technique for calibrating binary classifiers is Platt scaling, which fits a logistic regression model on the original outputs. A straightforward application of Platt scaling in our seq2seq setting computes the calibrated confidence score by $\sigma(w_t \cdot \log p_\theta(t \mid e) + b_t)$, where $\sigma$ is the sigmoid function and calibration parameters $w_t$ and $b$ are estimated on the development set by minimizing the binary cross-entropy loss.

Inspired by previous work (Zhao et al., 2021) which measures the bias of seq2seq models by feeding them with empty inputs, we propose to learn a weighted combination of both the conditional probability $p_\theta(t \mid e)$ and model bias $p_\theta(t \mid \varnothing)$. Specifically, we propose

$$\sigma \left( w_t^{(1)} \cdot \log p_\theta(t \mid e) + w_t^{(2)} \cdot \log p_\theta(t \mid \varnothing) + b_t \right)$$

as the calibrated confidence score. We will show in section 4 that incorporating the model bias term improves task performance and reduces calibration error.

**Multi-label Platt Scaling**: We now discuss the extension of this equation in the multi-label setting where $|\mathcal{T}| \gg 1$. A naive extension that considers each type independently would introduce $3|\mathcal{T}|$ parameters and involve training $|\mathcal{T}|$ logistic regression models on $|\mathcal{D}_{\text{dev}}| \cdot |\mathcal{T}|$ data points. To mitigate this difficulty, we propose to share calibration parameters across types based on their occurrence

**Algorithm 1:** Calibration parameters estimation

```
1  function GetCalibrationParams(𝒟dev, model,
     n_groups)
2      D ← [[] for i in range(n_groups)]
       // D stores the data points for
          estimating calibration parameters
3      for e, types in 𝒟dev do
4          for t in model.beam_search(e) do
5              X ← [log pθ(t|e), log pθ(t|∅)]
6              if t in types then
7                  y ← +1
8              else
9                  y ← -1
10             D[φ(t)].append((X, y))
11     W ← np.zeros((n_groups, 2))
12     B ← np.zeros(n_groups)
13     for i in range(n_groups) do
14         W[i, :], B[i] ←
             FitLogisticRegression(D[i])
15     return W, B
```

frequency in the dataset:

$$\hat{p}(t \mid e) = \sigma\Big(w^{(1)}_{\phi(t)} \cdot \log p_\theta(t \mid e)$$
$$+ w^{(2)}_{\phi(t)} \cdot \log p_\theta(t \mid \varnothing) + b_{\phi(t)}\Big) \quad (3)$$

where

$$\phi(t) = \big\lceil \log_2\big(\texttt{Freq}(t) + 1\big) \big\rceil \quad (4)$$

maps type $t$ to its frequency category.[5] Intuitively, rare types are more vulnerable to model bias thus should be handled differently compared to frequent types.

Furthermore, instead of training logistic regression models on all $|\mathcal{D}_{\text{dev}}| \cdot |\mathcal{T}|$ data points, we propose a sparse approximation strategy that only leverages candidate types generated by the seq2seq model via beam search.[6] This ensures that the entire calibration process retains the same time complexity as a regular evaluation run on the development set. The pseudo code for estimating calibration parameters is outlined in algorithm 1. Once the calibration parameters have been estimated, we select the optimal threshold by running a simple linear search.

### 3.5 Inference

At test time, given an entity mention $e$, we employ constrained beam search to generate a set of candidate types autoregressively. Following previous

work (De Cao et al., 2020, 2022), we pre-compute a prefix trie based on $\mathcal{T}$ and force the model to select valid tokens during each decoding step. Next, we compute the calibrated confidence scores using Equation 3 and discard types whose scores fall below the threshold.

In section 4, we also conduct experiments on single-label entity typing tasks. In such cases, we directly score each valid type using Equation 3 and select the type with the highest confidence score.

## 4 Experiments

### 4.1 Datasets

We use the UFET dataset (Choi et al., 2018), a standard benchmark for ultra-fine entity typing. This dataset contains 10331 entity types and is curated by sampling sentences from GigaWord (Parker et al., 2011), OntoNotes (Hovy et al., 2006) and web articles (Singh et al., 2012).

To test the out-of-domain generalization abilities of our model, we construct five entity typing datasets for three specialized domains. We derive these from existing NER datasets, WNUT2017 (Derczynski et al., 2017), JNLPBA (Collier and Kim, 2004), BC5CDR (Wei et al., 2016), MIT-restaurant and MIT-movie.[7] We treat each annotated entity mention span as an input to our entity typing model. WNUT2017 contains user-generated text from platforms such as Twitter and Reddit. JNLPBA and BC5CDR are both sourced from scientific papers from the biomedical field. MIT-restaurant and MIT-movie are customer review datasets from the restaurant and movie domains respectively. Table 1 provides the statistics and an example from each dataset.

### 4.2 Implementation

We initialize the seq2seq model with pretrained T5-large (Raffel et al., 2020) and finetune it on the UFET training set with a batch size of 8. We optimize the model using Adafactor (Shazeer and Stern, 2018) with a learning rate of 1e-5 and a constant learning rate schedule. The constrained beam search during calibration and inference uses a beam size of 24. We mark the entity mention span with a special token and format the input according to the template "{CONTEXT}  {ENTITY} is ". Input and the target entity type are tokenized using the standard T5 tokenizer.

---

[5]On the UFET dataset, this reduces the number of calibration parameters from 30993 to 27.

[6]This reduces the maximum number of calibration data points to $|\mathcal{D}_{\text{dev}}| \times$ BeamSize.

[7]https://groups.csail.mit.edu/sls/downloads/

| Dataset | Domain | Entity Types ($\mathcal{T}$) | Example |
|---------|--------|------------------------------|---------|
| UFET | News, web articles | 10331 types | [The explosions]event, calamity, attack, disaster occurred on the night of October 7, against the Hilton Taba and campsites used by Israelis in Ras al-Shitan. |
| WNUT2017 | Social media | {corporation, creative_work, group, location, person, product} | RT @MarshmallowDoof: I did drawn the [Tiger Mama]creative_work @BuxbiArts |
| JNLPBA | Biomedical | {DNA, RNA, cell_line, cell_type, protein} | In vivo control of [NF-kappa B]protein activation by I kappa B alpha. |
| BC5CDR | Biomedical | {disease, chemical} | In a previous phase II study with 3 - weekly bolus [5-FU]chemical, FA and mitomycin C ( MMC ) we found a low toxicity rate and response rates comparable to those of regimens such as ELF, FAM or FAMTX, and a promising median overall survival. |
| MIT-restaurant | Customer review | {rating, amenity, location, restaurant, price, hours, dish, cuisine} | Can you make a reservation at [pf changs]restaurant for tonight? |
| MIT-movie | Customer review | {actor, plot, opinion, award, year, genre, origin, director, soundtrack, relationship, character, quote} | An [animated]genre movie about a criminal mastermind that attempts to steal the moon |

Table 1: Dataset statistics and examples. Only UFET has multiple types for each entity mention.

## 4.3 Baselines

We compare our method to previous state-of-the-art approaches, including multi-label classifier-based methods such as BiLSTM (Choi et al., 2018), BERT, Box4Types (Onoe et al., 2021) and ML-MET (Dai et al., 2021). In addition, we include a bi-encoder model, UniST (Huang et al., 2022) as well as the current state-of-the-art method, LITE (Li et al., 2022), which is based on a cross-encoder architecture.

We also compare with ChatGPT [8] and Flan-T5-XXL (Chung et al., 2022), two large language models that have demonstrated impressive few-shot and zero-shot performance across various tasks. For the UFET dataset, we randomly select a small set of examples from the training set as demonstrations for each test instance. Instruction is provided before the demonstration examples to facilitate zero-shot evaluation. Furthermore, for the five cross-domain entity typing datasets, we supply ChatGPT and Flan-T5-XXL with the complete list of valid types. Sample prompts are shown in Appendix A.

## 5 Results

### 5.1 UFET

In Table 2, we compare our approach with a suite of baselines and state-of-the-art systems on the UFET dataset. Our approach outperforms LITE (Li et al., 2022), the current leading system based on a cross-encoder architecture, with a 0.7% improvement in

---

[8]We use the `gpt-3.5-turbo-0301` model available via the OpenAI API.

---

| Method | P | R | F1 |
|--------|---|---|-----|
| *Few-shot methods* | | | |
| ChatGPT (0-shot) | 55.5 | 10.5 | 17.6 |
| ChatGPT (8-shot) | 46.7 | 34.9 | 40.0 |
| ChatGPT (16-shot) | 47.8 | 36.7 | 41.5 |
| ChatGPT (32-shot) | 45.9 | 37.3 | 41.2 |
| *Supervised methods* | | | |
| BiLSTM (Choi et al., 2018) | 47.1 | 24.2 | 32.0 |
| BERT (Onoe and Durrett, 2019) | 51.6 | 33.0 | 40.2 |
| Box4Types (Onoe et al., 2021) | 52.8 | 38.8 | 44.8 |
| MLMET (Dai et al., 2021) | 53.6 | 45.3 | 49.1 |
| UniST (Huang et al., 2022) | 50.2 | 49.6 | 49.9 |
| LITE (Li et al., 2022) | 52.4 | 48.9 | 50.6 |
| CASENT (Ours) | 53.3 | 49.5 | **51.3** |

Table 2: Macro-averaged precision, recall and F1 score (%) on the UFET test set. The model with highest F1 score is shown in **bold** and the second best is underlined.

the F1 score. Among the fully-supervised models, cross-encoder models demonstrate superior performance over both bi-encoder methods and multi-label classifier-based models.

ChatGPT exhibits poor zero-shot performance with significantly low recall. However, it is able to achieve comparable performance to a BERT-based classifier with a mere 8 few-shot examples. Despite this, its performance still lags behind recent fully supervised models.

### 5.2 Out-of-domain Generalization

We evaluate the out-of-domain generalization performance of different models on the five datasets discussed in §4.1. The results are presented in Table 3. It is important to note that we don't compare

| Method | Social Media | Biomedical | | Customer Review | | |
| | WNUT 2017 | JNLPBA | BC5CDR | MIT-restaurant | MIT-movie | Avg. |
| --- | --- | --- | --- | --- | --- | --- |
| *Zero-shot methods* | | | | | | |
| Random | 16.7 | 20.0 | 50.0 | 12.5 | 8.3 | 21.5 |
| Flan-T5-XXL | 62.9 | 71.8 | 63.0 | 39.6 | 45.4 | 56.5 |
| ChatGPT | 76.3 | 85.4 | 96.7 | 80.3 | 77.2 | 83.2 |
| LITE (Li et al. 2022) | 67.0 | 74.9 | 96.1 | 47.7 | 54.5 | 68.0 |
| CASENT (no finetuning, no calibration) | 65.5 | 79.2 | 98.2 | 52.9 | 51.2 | 69.4 |
| *Few-shot methods* | | | | | | |
| RoBERTa-large (finetuned on 50 examples) | 65.5 | 85.1 | 96.2 | 75.0 | 69.9 | 78.3 |
| CASENT (no finetuning, calibration on dev) | 74.2 | 84.0 | 98.2 | 68.5 | 71.7 | 79.3 |
| CASENT (finetuned on 50 examples) | **77.3** | **92.2** | **98.8** | **81.8** | **86.2** | **87.2** |

Table 3: Test set accuracy on five specialized domain entity typing datasets derived from existing NER datasets. The best score is shown in **bold** and the second best is underlined. The results of LITE are obtained by running inference using the model checkpoint provided by the authors.

with multi-label classifier models like Box4Types and MLMET that treat types as integer indices, as they are unable to generalize to unseen types.

In the zero-shot setting, LITE and CASENT are trained on the UFET dataset and directly evaluated on the target test set. Flan-T5-XXL and ChatGPT are evaluated by formulating the task as a classification problem with all valid types as candidates. As shown in Table 3, ChatGPT demonstrates superior performance with a large margin compared to other models. This highlights ChatGPT's capabilities on classification tasks with a small label space. Our approach achieves comparable results to LITE and significantly outperforms Flan-T5-XXL, despite having less than $10\%$ of its parameters.

We also conduct experiments in the few-shot setting, where either a small training set or development set is available. We first explore re-estimating the calibration parameters of CASENT on the target development set by following the process discussed in §3.4 without weight sharing and sparse approximation.[9] Remarkably, this re-calibration process, without any finetuning, results in an absolute improvement of $+9.9\%$ and comparable performance with ChatGPT on three out of five datasets. When finetuned on 50 randomly sampled examples, our approach outperforms ChatGPT and a finetuned RoBERTa model by a significant margin, highlighting the benefits of transfer learning from the ultra-fine entity typing task.

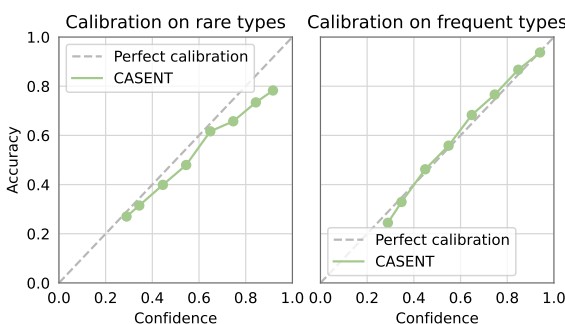

Figure 3: Reliability diagrams of CASENT on the UFET test set. The left diagram represents rare types with fewer than 10 occurrences while the right diagram represents frequent types.

## 6 Analysis

### 6.1 Calibration

Table 4 presents the calibration error of different approaches. We report Expected Calibration Error (ECE) and Total Calibration Error (TCE) which measures the deviation of predicted confidence scores from empirical accuracy. Interestingly, we observe that the entailment scores produced by LITE, the state-of-the-art cross-encoder model, are poorly calibrated. Our approach achieves slightly lower calibration error than Box4Types, which applies temperature scaling (Guo et al., 2017) to the output of a BERT-based classifier. Figure 3 displays the reliability diagrams of CASENT for both rare types and frequent types. As illustrated by the curve in the left figure, high-confidence predictions for rare types are less well-calibrated.

---

[9]The number of calibration parameters is $3|\mathcal{T}|$, which is less than 40 on all five datasets.

| Method | Calibration Method | Test-F1 (%) | Test-ECE (%) | Test-TCE (%) | Dev-TCE (%) |
|---|---|---|---|---|---|
| Box4Types | Temperature scaling | 44.8 | - | - | 11.19 |
| LITE | - | 50.6 | 52.36 | 52.36 | 52.56 |
| CASENT | Eq. 3 | **51.3** | **1.23** | **9.75** | **9.38** |
| | Eq. 3 without the model bias term $p_\theta(t \mid \varnothing)$ | 49.4 | 3.87 | 20.34 | 14.76 |
| | Eq. 3 with $\phi(t) = t$ (independent weights) | 48.8 | 7.37 | 57.00 | 9.72 |
| | Eq. 3 with $\phi(t) = t_0$ (all types share same weights) | 47.8 | 3.89 | 34.57 | 36.29 |
| | $p_\theta(t \mid e)$ (no calibration) | 47.3 | 12.19 | 118.16 | 100.31 |

Table 4: Macro F1, ECE (Expected Calibration Error) and TCE (Total Calibration Error) on the UFET dataset. ECE and TCE are computed using 10 bins. The best score is shown in **bold**. Onoe et al. (2021) only reported calibration results on the dev set thus the results of Box4Types on the test set are not included.

| Method | # params | F1 |
|---|---|---|
| T5-small | 80M | 40.9 |
| T5-small + CASENT | | 47.2 |
| T5-base | 250M | 45.4 |
| T5-base + CASENT | | 49.6 |
| T5-large | 780M | 47.3 |
| T5-large + CASENT | | 51.3 |
| T5-3B | 3B | 48.6 |
| T5-3B + CASENT | | 51.4 |

Table 5: Macro F1 score (%) of CASENT on the UFET test set with different T5 variants.

| Method | Training Time | Inference Latency | GPU Mem. |
|---|---|---|---|
| MLMET | 180h[†] | $0.02 \pm 0.05$s | 0.5Gb |
| LITE | 40h[†] | $23.1 \pm 5.73$s | 1.4Gb |
| CASENT | 6h | $0.39 \pm 0.04$s | 2.8Gb |

Table 6: Training time, inference latency and inference time GPU memory usage estimated on a single NVIDIA RTX A6000 GPU. Inference time statistics are estimated using 100 random UFET examples. Results marked by † are reported by Li et al. (2022).

## 6.2 Ablation Study

We also perform an ablation study to investigate the impacts of various design choices in our proposed calibration method. Table 4 displays the results of different variants of CASENT. A vanilla seq2seq model without any calibration yields both low task performance and high calibration error, highlighting the importance of calibration. Notably, a naive extension of Platt scaling that considers each type independently leads to significant overfitting, illustrated by an absolute difference of $47.28\%$ TCE between the development and test sets. Removing the model bias term also has a negative impact on both task performance and calibration error.

## 6.3 Choice of Seq2seq Model

In Table 5, we demonstrate the impact of calibration on various T5 variants. Our proposed calibration method consistently brings improvement across models ranging from 80M parameters to 3B parameters. The most substantial improvement is achieved with the smallest T5 model.

## 6.4 Training and Inference Efficiency

In Table 6, we compare the efficiency of our method with previous state-of-the-art systems. Remarkably, CASENT only takes 6 hours to train on a

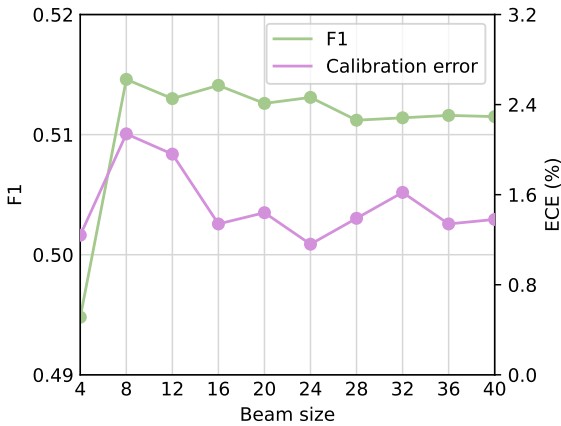

Figure 4: Test set Macro F1 score and Expected Calibration Error (ECE) with respect to the beam size on the UFET dataset.

single GPU, while previous methods require more than 40 hours. While CASENT achieves an inference speedup of over 50 times over LITE, it is still considerably slower than MLMET, a BERT-based classifier model. This can be attributed to the need for autoregressive decoding in CASENT.

## 6.5 Impact of Beam Size

Given that the inference process of CASENT relies on constrained beam search, we also investigate the impact of beam size on task performance and calibration error. As shown in Figure 4, a beam

size of 4 results in a low calibration error but also low F1 scores, as it limits the maximum number of predictions. CASENT consistently maintains high F1 scores with minor fluctuations for beam sizes ranging from 8 to 40. On the other hand, a beam size between 8 and 12 leads to high calibration errors. This can be attributed to our calibration parameter estimation process in algorithm 1, which approximates the full $|\mathcal{D}_{\text{dev}}| \cdot |\mathcal{T}|$ calibration data points using model predictions generated by beam search. A smaller beam size leads to a smaller number of calibration data points, resulting in a suboptimal estimation of calibration parameters.

## 7  Conclusion

Engineering decisions often involve a tradeoff between efficiency and accuracy. CASENT simultaneously improves upon the state-of-the-art in both dimensions while also being conceptually elegant. The heart of this innovation is a constrained beam search with a novel probability calibration method designed for seq2seq models in the multi-label classification setting. Not only does this method outperform previous methods—including ChatGPT and the existing fully-supervised methods—on ultrafine entity typing, but it also exhibits strong generalization capabilities to unseen domains.

## 8  Limitations

While our proposed CASENT model shows promising results on ultra-fine entity typing tasks, it does have certain limitations. Our experiments were conducted using English language data exclusively and it remains unclear how well our model would perform on data from other languages. In addition, our model is trained on the UFET dataset, which only includes entity mentions that are identified as noun phrases by a constituency parser. Consequently, certain types of entity mentions such as song titles are excluded. The performance and applicability of our model might be affected when dealing with such types of entity mentions. Future work is needed to adapt and evaluate the proposed approach in other languages and broader scenarios.

## Acknowledgements

This material is based on research sponsored by the Air Force Research Laboratory under agreement number FA8750-19-2-0200. The U.S. Government is authorized to reproduce and distribute reprints for Governmental purposes notwithstanding any copyright notation thereon. The views and conclusions contained herein are those of the authors and should not be interpreted as necessarily representing the official policies or endorsements, either expressed or implied, of the Air Force Research Laboratory or the U.S. Government.

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

## A  ChatGPT / Flan-T5 Prompts

Below is a sample prompt for ChatGPT and Flan-T5-XXL for the five out-of-domain datasets:

> Instruction: Identify the type of the entity mention tagged by . Output the type directly and do not write any explanation.
> Choices: DNA, RNA, cell_line, cell_type, protein
> Entity: Number of glucocorticoid receptors in lymphocytes and their sensitivity to hormone action .
> Label:

For the UFET dataset, it is not feasible to provide the model with the entire type vocabulary. Instead we provides demonstration examples sampled from the training set. Below is a sample prompt with two demonstration examples:

```
Instruction:   Predict   the   fine-grained
entity types for the entity mention tagged
by . Separate the types with commas.

Entity:   He  get  's  zero
from Arafat , " said Benjamin Begin , the
science minister .
Labels: academician, scientist, person

Entity:   President   Obama   's   surprise
proposal  to  cancel  the  $  108  billion
moon program and the jobs that go with
it  triggered  an  uproar  in
Texas ,  Florida  and  other  states  with
space - related industries .
Labels: work, job, bill

Entity: On late Monday night ,
30th Nov 2009 , Bangladesh Police arrested
Rajkhowa somewhere near Dhaka .
Labels:
```