# OpenReview forum: "Calibrated Seq2seq Models for Efficient and Generalizable Ultra-fine Entity Typing"
_EMNLP/2023/Conference — EMNLP 2023 Findings_

### Official Review · Reviewer_NPCS · 2023-08-01

**Typos Grammar Style And Presentation Improvements:** line 	492
**Soundness:** 3

**Excitement:**

3: Ambivalent: It has merits (e.g., it reports state-of-the-art results, the idea is nice), but there are key weaknesses (e.g., it describes incremental work), and it can significantly benefit from another round of revision. However, I won't object to accepting it if my co-reviewers champion it.

**Paper Topic And Main Contributions:**

Authors claim a new seq2seq model, CASENT, for ultra-fine entitiy typing. The core contribution is a calibration method that transforms conditional log-probability into calibrated confidence value. Authors also show very positive experiment results.

**Questions For The Authors:**

1. Is the \mathcal{T} the types used in Choi’s work?
2. In Figure 1, the input of calibration is person, criminal, adolescent, …, where is the adolescent after the calibration?
3. Why after probability calibration, the sum of calibrated values are greater than 1?

**Reasons To Accept:**

The experiment results are very good.

**Reasons To Reject:**

The paper has not much novelty. The definition of Ultra-fine Entity Typing in this paper is not exactly the same of the original definition. In the original definition of Ultra-Fine Entity Typing by Choi, et al. (2018), there is neither restriction about the number of types, nor restriction about the form. In the task definition in Section 3.1, types are restricted to the set of \mathcal{T}. Although it is the types used in Choi’s work, fixed type  structure is not what Ultra-Fine entity typing means.

**Reproducibility:**

2: Would be hard pressed to reproduce the results. The contribution depends on data that are simply not available outside the author's institution or consortium; not enough details are provided.

**Reviewer Confidence:**

3: Pretty sure, but there's a chance I missed something. Although I have a good feel for this area in general, I did not carefully check the paper's details, e.g., the math, experimental design, or novelty.

---

> ### Author Rebuttal · Authors · 2023-08-28
>
> >  The definition of Ultra-fine Entity Typing in this paper is not exactly the same of the original definition. In the original definition of Ultra-Fine Entity Typing by Choi, et al. (2018), there is neither restriction about the number of types, nor restriction about the form.
>
> - We follow the standard evaluation protocol of UFET defined by Choi et al. 2018.
> - All UFET methods in Table 1, including Choi et. al 2018, use the same 10k type vocabulary (see Sec 2.1-2.2 of the Choi et al. paper).
>
> Below we compare CASENT (our method) with prior methods  regarding the restriction on \mathcal{T}:
> - Methods based on multi-label classifiers (e.g., Box4Types, MLMET, …) can only predict types in the dataset’s \mathcal{T}
> - LITE and CASENT can generalize to types outside of the dataset’s \mathcal{T}. This is illustrated in our out-of-domain experiments (Table 3).
> - CASENT can be easily extended to predict free-form types by replacing the constrained beam search with standard beam search. In contrast, LITE requires a predefined list of candidate types due to their formulation as an NLI task.
>
> > Is the \mathcal{T} the types used in Choi’s work?
>
> Yes
>
> > In Figure 1, the input of calibration is person, criminal, adolescent, …, where is the adolescent after the calibration?
>
> “adolescent” still exists but is ranked lower (with a confidence score of ~0.1). Many types are omitted in Fig. 1 due to space limitations.
>
> >Why after probability calibration, the sum of calibrated values are greater than 1?
>
> UFET is a multi-label classification task where each entity can map to multiple types. Therefore, unlike the single-label multi-class classification, label probabilities don’t need to add up to 1.
>
> >The paper has not much novelty.
>
> We would like to highlight the novelty of our model with the following points:
> - CASENT goes beyond heuristics for adjusting raw probabilities; rather, it is a principled calibration method that is generally applicable to any multi-label classification task (such as UFET). CASENT achieves not only SOTA task performance (measured by F1) but also SOTA calibration error (measured by ECE/TCE). This is also acknowledged by Reviwere rRPt.
> - Probability calibration is a core topic in ML studied by well-known papers (e.g., Guo et al., 2017) and requires non-trivial methods beyond post-hoc adjustments (see Sec. 2.2). To our knowledge, CASENT is the first calibration algorithm proposed for extreme multi-label classification tasks with thousands of classes like UFET. Results in Table 3 show that our proposed calibration algorithm reduces the expected calibration error (ECE) from 12.19 to 1.16.
> - CASENT is the first model that offers 1) generalization to unseen types, 2) efficient training & inference, 3) calibrated confidence scores (see lines 82-90, Sec.1). This fundamentally distinguishes CASENT from all previous approaches.
>
> |                      | Efficient training & inference | Generalization to unseen types | Calibrated confident scores |
> | -------------------- | ------------------------------ | ------------------------------ | --------------------------- |
> | Box4Types, MLMET, …  |Yes                              |                  No              |         No (except Box4types)                    |
> | LITE (previous SOTA) |  No                              | Yes                              |   No                          |
> | CASENT               | Yes                              | Yes                              |Yes                          |

---

### Official Review · Reviewer_hY4H · 2023-08-02

**Soundness:** 3

**Excitement:**

2: Mediocre: This paper makes marginal contributions (vs non-contemporaneous work), so I would rather not see it in the conference.

**Paper Topic And Main Contributions:**

This paper proposes a calibration process on a seq2seq model trained for fine-grained entity typing. Specifically, after the model is trained to generate fine-grained types in the output sequence, the authors propose to calibrate the produced label confidences during inference time. The calibration process essentially learns a regression model using the annotated labels of the dev set that addresses some label prior biases.

Experiments show that the proposed seq2seq entity typing model after the proposed calibration process outperforms baseline models, and the trained model can generalize to out-of-domain typing benchmarks with a few training examples.

**Reasons To Accept:**

The proposed system has good performances, and the calibration process seems effective on out-of-domain datasets.

**Reasons To Reject:**

1) The experiment setting needs improvements. First of all, I think the authors should add the baseline seq2seq model (T5 in this case) to all experiments for us to understand better how well the calibration process is working. The key ablation experiment in Table 4 seems flawed because the proposed calibration process uses more gold-labeled instances (the dev set), so the baseline T5 model should be trained on a joint set combining the training data and the dev data.

2) The contribution is limited as a long paper. In my view, the only contribution of this work is the calibration process that employs some tricks to address label prior biases from the occurring frequencies. Much content in the current version (e.g., Fig.1, Table 1, even the ablation studies in Table 4 because it can be simply combined if T5 is used as a baseline in main experiments) is unnecessary.

**Reproducibility:**

4: Could mostly reproduce the results, but there may be some variation because of sample variance or minor variations in their interpretation of the protocol or method.

**Reviewer Confidence:**

4: Quite sure. I tried to check the important points carefully. It's unlikely, though conceivable, that I missed something that should affect my ratings.

---

> ### Author Rebuttal · Authors · 2023-08-28
>
> We thank the reviewer for their constructive comments.
> > First of all, I think the authors should add the baseline seq2seq model (T5 in this case) to all experiments for us to understand better how well the calibration process is working.
>
> The baseline seq2seq model (T5 w/o calibration) has been included in both Table 4 (last row) and Table 5 (all rows).
> - Results in Table 4 show that CASENT achieves significant improvements over baseline T5 in terms of both F1 score and calibration error (ECE, TCE)
> - Results in Table 5 show that CASENT achieves significant improvements over all T5 variants from T5-small to T5-3B.
>
> > The key ablation experiment in Table 4 seems flawed because the proposed calibration process uses more gold-labeled instances (the dev set), so the baseline T5 model should be trained on a joint set combining the training data and the dev data.
>
> This is an interesting issue, and one which we also considered. Here is our reasoning for why we chose the experimental settings that we did:
> - Using dev sets for calibration and hyperparameter estimation is a standard practice in the ML literature. Calibration parameters function similarly as hyperparameters as they can only be estimated after the model has been trained.
> - Box4types (Onoe et al., 2021) follows the same setting. They trained BERT on the training set and did calibration on the dev set.
> - We followed the standard practice and used the dev set to estimate 27 calibration parameters and 1 threshold. We didn’t use the dev set to update any of the 770M T5 parameters. We ensure that all major results are based on the unseen test set.
> - All UFET models, including baseline T5, require the dev set for hyperparameter tuning and model selection (which involves at least tuning the threshold on the dev set). Therefore it is unreasonable to use dev set for training.
>
> To address the concern raised by the reviewer that using dev set for calibration might account for the major improvement of CASENT over baseline T5, we would like to present the following two new results:
> - The dev set F1 score of CASENT is 51.5 (test F1: 51.3), which shows that CASENT does not overfit the dev set.
> - We trained baseline T5 on train+dev as suggested by the reviewer, showing that CASENT (without training on dev) still significantly outperforms it. Training CASENT on train+dev similarly yields (a marginal) improvement.
> | Method                      | Training data | UFET Test F1 |
> | --------------------------- | ------------- | ------------ |
> | Baseline T5 w/o calibration | train         | 47.3         |
> | Baseline T5 w/o calibration | train+dev     | 47.7         |
> | CASENT                      | train         | 51.3         |
> | CASENT                      | train+dev     | 51.5         |
>
> > In my view, the only contribution of this work is the calibration process that employs some tricks to address label prior biases from the occurring frequencies.
> We would like to highlight the novelty of our model with the following points:
> - CASENT goes beyond heuristics for adjusting raw probabilities; rather, it is a principled calibration method that is generally applicable to any multi-label classification task (such as UFET). CASENT achieves not only SOTA task performance (measured by F1) but also SOTA calibration error (measured by ECE/TCE). This is also acknowledged by Reviwere rRPt.
> - Probability calibration is a core topic in ML studied by well-known papers (e.g., Guo et al., 2017) and requires non-trivial methods beyond post-hoc adjustments (see Sec. 2.2). To our knowledge, CASENT is the first calibration algorithm proposed for extreme multi-label classification tasks with thousands of classes like UFET. Results in Table 3 show that our proposed calibration algorithm reduces the expected calibration error (ECE) from 12.19 to 1.16.
> - CASENT is the first model that offers 1) generalization to unseen types, 2) efficient training & inference, 3) calibrated confidence scores (see lines 82-90, Sec.1). This fundamentally distinguishes CASENT from all previous approaches.
>
> |                      | Efficient training & inference | Generalization to unseen types | Calibrated confident scores |
> | -------------------- | ------------------------------ | ------------------------------ | --------------------------- |
> | Box4Types, MLMET, …  |Yes                              |                  No              |         No (except Box4types)                    |
> | LITE (previous SOTA) |  No                              | Yes                              |   No                          |
> | CASENT               | Yes                              | Yes                              |Yes                          |
>
>
> > ablation studies in Table 4 are unnecessary because they can be simply combined if T5 is used as a baseline in main experiments
>
> Ablation studies in table 4 demonstrate the following two points (discussed in Sec. 6.1 - 6.2) that can be not be derived from the main results:
> - Various design choices of CASENT are reasonable and empirically useful.
> - CASENT archives state-of-the-art calibration error.

---

### Official Review · Reviewer_rRPt · 2023-08-05

**Soundness:** 4

**Excitement:**

4: Strong: This paper deepens the understanding of some phenomenon or lowers the barriers to an existing research direction.

**Paper Topic And Main Contributions:**

This paper proposes a novel entity typing model (CASENT) that uses a calibration module to adjust the raw output. During inference, CASENT first employs a seq2seq model (e.g., T5) and autoregressively predicts entity types given an entity mention by using constrained beam search (i.e., forcing the model to select only predefined entity types). Then, the calibration module, which uses logistic regression with three parameters, recomputes the final probability based on the raw probability and the probability of an entity type given an empty input. To reduce the total number of parameters in the calibration module, the authors bucket the candidate entity types by frequency in the data and share the logistic regression parameters within each bucket. Additionally, this calibration module is trained with entity types existing in the beam, rather than with |T| types, to manage time complexity.

This approach is evaluated using entity typing benchmarks such as UFET, and is compared against SOTA approaches, including ChatGPT. The main result is observed on UFET, where CASENT outperforms all baselines, showing the best calibration errors. In the zero-shot setting, models are trained on UFET and evaluated on other entity typing benchmarks. In this setting, ChatGPT demonstrates strong results, indicating that the generalization abilities of smaller models are limited, while calibration enhances performance.


**Questions For The Authors:**

- How did you prompt ChatGPT? It would be helpful to include a discussion about its sensitivity to various prompts, etc.

**Reasons To Accept:**

- CASENT includes architectural contributions for ultra-fine entity typing such as multi-label Platt scaling and an efficient training method. Each design choice is well-motivated and reasonable.
- Experimental results show that the proposed approach is effective, outperforming prior work (the margin is 0.7 but it’s not small given that UFET is 10k multi-label classification).
- I really appreciate that the authors compare their approaches with SOTA blackbox LLMs such as ChatGPT, confirming that supervised methods still exceed ChatGPT on ultra-fine entity typing.
- This paper is well written and easy to read. All key concepts are clearly explained.


**Reasons To Reject:**

- The evaluation focuses on quantitative results. Without qualitative analysis, it’s hard to see when/why calibration helps.
- Similarly, no error analysis on baseline outputs is included. It’s difficult to understand what types of errors have been fixed by the proposed approach.


**Reproducibility:**

4: Could mostly reproduce the results, but there may be some variation because of sample variance or minor variations in their interpretation of the protocol or method.

**Reviewer Confidence:**

5: Positive that my evaluation is correct. I read the paper very carefully and I am very familiar with related work.

---

> ### Author Rebuttal · Authors · 2023-08-28
>
> We thank the reviewer for their constructive comments.
>
> > The evaluation focuses on quantitative results. Without qualitative analysis, it’s hard to see when/why calibration helps.
> Similarly, no error analysis on baseline outputs is included. It’s difficult to understand what types of errors have been fixed by the proposed approach.
>
> Thank you for making these excellent points. We have included an example in Fig. 1 but found it difficult to include discussion of our qualitative evaluation or error analysis due to space but acknowledge that it would greatly enhance the significance of our work. In the final version of the paper, we will use some of the additional space to present a richer analysis of the results errors and how they compare to the baseline outputs.
>
> Here are three examples: the first is from Fig. 1 with added baseline outputs, and the others are randomly sampled. The comparison between CASENT and T5 (CASENT w/o calibration) demonstrates that calibration reduces hallucination by reranking irrelevant concepts with high raw probabilities. Calibration also produces more consistent confidence scores (e.g. baseline T5 assigns higher confidence to “agency” than “organization” while “agency” is a subtype of “organization”).
> *****
> Example 1
> *****
> Entity: A court in Jerusalem sentenced <M>a Palestinian</M> to 16 life terms for forcing a bus off a cliff July 6, killing 16 people, Israeli radio reported.
>
> CASENT: person (0.98) criminal (0.65) male (0.55)
>
> Baseline T5: person (0.61), criminal (0.38), male (0.30), adolescent (0.25), convict (0.25)
>
> Gold types: person, criminal
> *****
> Example 2
> *****
> Entity: She was made a doctor of music by the University of Cambridge in 1976 , and became <M>a Dame Commander of the Order of the British Empire -LRB- DBE -RRB-</M> in 1992 .
>
> CASENT: person (0.96), honor (0.45), aristocrat (0.36), leader (0.30), adolescent (0.29), honorary (0.28)
>
> Baseline T5: person (0.51), adolescent (0.41), aristocrat (0.38), honor (0.36), adversity (0.27)
>
> Gold types: person, occupation, title
> *****
> Example 3
> *****
> Entity: NASA said the Galileo worm hadn't affected its computers or the computers of other government agencies because <M>they</M> had modified their systems to reject worms.
>
> CASENT: organization (0.67), agency (0.67), government (0.57), administration (0.48), authority (0.33), corporation (0.29)
>
> Baseline T5: agency (0.46), organization (0.40), bureaucracy (0.35), government (0.33), administration (0.28)
>
> Gold types: organization, administration, group, working group
>
> > How did you prompt ChatGPT? It would be helpful to include a discussion about its sensitivity to various prompts, etc.
>
> Flan-T5-XXL/ChatGPT prompt for UFET:
> ```
> Instruction: Predict the fine-grained entity types for the entity mention tagged by <mark>. Separate the types with commas.
> {randomly sampled demonstration examples}
> Entity: {entity_mention}
> Labels:
> ```
> Flan-T5-XXL/ChatGPT prompt for the five specialized domain datasets:
> ```
> Instruction: Identify the type of the entity mention tagged by <mark>. Output the type directly and do not write any explanation.
> Choices: {candidate_types}
> Entity: {entity_mention}
> Label:
> ```
> We manually refine the instruction part of the prompt for both datasets until ChatGPT’s outputs do not have formatting issues. We also experiment with different formats of the entity mention (e.g. listing the context and the entity mention span separately, or tagging the entity mention within the context) and find that ChatGPT’s performance is quite robust (< 2% accuracy difference).

---

### Meta-Review · Area_Chair_U8Vj · 2023-09-20

**Recommendation:** 3

**Metareview:**

This paper proposes a calibration method on a seq2seq model for fine-grained entity typing, which can give label confidence during inference time after the model is trained to generate fine-grained types. The main idea is to transform conditional log-probability into a calibrated confidence value, which shows promising results in experiments. Although some reviewers are concerned about the novelty and contribution of this paper, the paper is well-organized and clearly stated.

---

### Decision · Program_Chairs · 2023-10-07

**Decision:**

Accept-Findings

**Comment:**

This paper proposes a calibration method on a seq2seq model for fine-grained entity typing, which can give label confidence during inference time after the model is trained to generate fine-grained types. The main idea is to transform conditional log-probability into a calibrated confidence value, which shows promising results in experiments. Although some reviewers are concerned about the novelty and contribution of this paper, the paper is well-organized and clearly stated.